# Prevalence of myopia among disadvantaged Australian schoolchildren: A 5-year cross-sectional study

Aicun Fu[1], Kathleen Watt[2], Barbara M. Junghans[2], Androniki Delaveris[2], Fiona Stapleton[2]*

1 Department of Ophthalmology, The First Affiliated Hospital of Zhengzhou University, Zhengzhou, China,
2 School of Optometry and Vision Science, UNSW, Sydney, Australia

* f.stapleton@unsw.edu.au

## Abstract

### Purpose

Myopia prevalence is influenced by environmental factors including heritability and social disadvantage. The current prevalence of myopia among disadvantaged school children in Australia has not been reported. Therefore, this study analyses refractive data for children from rural and outer suburban areas.

### Methods

The records of 4,365 children aged 6–15 visiting a city-based government-school respite care center during the years 2014/2016/2018 were analyzed for right eye non-cycloplegic spherical equivalent refraction (SER). The prevalence of myopia (SER≤-0.50D) was compared with historical data.

### Results

The prevalence of myopia was 3.5%, 4.4% and 4.3% in 2014, 2016 and 2018, respectively. The prevalence of myopia increased with age (P<0.0001), but was not related to sex or year of testing (all P >0.05). The overall mean SER was 0.89±0.86D, 0.62±0.89D and 0.56±0.95 in 2014, 2016 and 2018, respectively. Mean SER was associated with year of testing, age (all P <0.0001) and sex (P = 0.03). Mean SER decreased slightly from 2014 to 2018 and demonstrated a significant shift towards less hyperopia with increasing age. Mean SER of females was higher than that of males and decreased faster than in males with age (P interaction = 0.03).

### Conclusions

Myopia prevalence increased with age. The mean SER decreased slightly from 2014 to 2018. Sex differences in the rate of change with age was observed. Compared with 40 years ago, the prevalence of myopia has doubled, but it remains significantly lower than in

**Data Availability Statement:** All relevant data are within the manuscript and its Supporting Information files.

**Funding:** Aicun Fu: Henan overseas study project of health and family planning technology

(2018038) NO-The funders had no role in study design, data collection and analysis, decision to publish, or preparation of the manuscript.

**Competing interests:** The authors have declared that no competing interests exist.

school children of a similar age living in established urban areas that are regarded as having a higher socioeconomic status.

## Introduction

There is evidence that myopia is reaching epidemic proportions at a speed that suggests strong environmental influences [1]. Myopia is not just an optical inconvenience, its detrimental impacts on the eyeball itself include a several fold increase in the risk of many sight-threatening ocular conditions, including cataracts, glaucoma, retinal detachments and myopic retinopathy [2].

There is clear evidence for a high and increasing prevalence of myopia in East Asia [3–8], apparently driven by increasing educational pressures and urbanization [9–13]. The absolute prevalence of myopia in Australia is much lower than in East Asian [14–16], United States [13] and Western Europe [17–20]. However, the evidence for an increasing prevalence of myopia in Australia is more questionable, due to the limited number of cross-sectional studies over time and the confounding effects of age-related emmetropization [21]. Therefore, studies at different time points are clearly required to quantify changes in the prevalence of myopia, for accurate assessment of the public health impacts and to assist with the development of preventive approaches.

There have been two cross-sectional studies at different time points evaluating the prevalence of myopia among schoolchildren in Sydney urban areas. French et al. [14] found that the prevalence of myopia in children aged 12 years living in Sydney over a 6-year period (from 2004–2005 to 2009–2011) increased from 4.4% to 8.4% in Caucasian and from 38.5% to 42.7% in East Asian children, respectively. Conversely, Junghans et al. [16] found little evidence of an epidemic of myopia in Australian primary school children aged 4–12 over a 30-year period, although they mention that only 8.8% of their cohort comprised Asian children.

Many studies have shown that the prevalence of myopia is influenced by environmental factors, such as living in urban as against rural areas [22–24], inner city-urban and outer suburban areas [25], population density and house size [26], and housing type [27]. Other external factors such as social disadvantage appear to impact myopia prevalence, such that children of parents having fewer years education and lower incomes are less likely to become myopic [28–30].

The Stewart House Children's Charity is located on the northern beaches in Sydney and provides a unique service for children coming from disadvantaged populations by offering respite care with health screening and personal health and wellbeing education and support [31]. Children attend the service from predominantly rural or remote areas of New South Wales and the Australian Capital Territory. The School of Optometry and Vision Science at the University of New South Wales (UNSW) has provided a comprehensive vision care service to Stewart House since 1972. This retrospective cross-sectional study captured 3 time points to assess the prevalence of myopia and mean spherical equivalent refraction (SER) among this population of disadvantaged schoolchildren in Sydney, Australia in 2014, 2016 and 2018. In addition, the data from this study was compared with historical data from the same location.

## Methods

This is a retrospective study of existing clinical records obtained during the Stewart House visits. The participants were from government schools in New South Wales and the Australian

Capital Territory. Each school Principal chose children from disadvantaged families to attend a 2-week program at Stewart House. An information flyer describing the Stewart House program and age-appropriate eye examination was sent to parents or guardians prior to the children attending the service, which required signed consent. A short questionnaire asking parents of their children if they had visual symptoms was included. All children who were granted permission by their parents or guardians to visit the Stewart House in 2014, 2016 and 2018 were included, regardless of visual status. Each child attended Stewart House once. This study was approved by the Human Ethics Committee of UNSW Sydney (HC No:190255) and conformed to the tenets of the Declaration of Helsinki. The Research Ethics Committee specifically waived the requirement for parental or guardian consent.

Refractive error was determined by noncycloplegic retinoscopy while the child maintained fixation on a distant non-accommodative (6 m) target. All retinoscopy was performed by one examiner (AD). Other tests included in the eye examination were letter visual acuity at 6 m and 33 cm, cover test for strabismus, ocular motilities, saccades, pupil reactions, near point of convergence, heterophoria, stereopsis, accommodative facility, color vision and ophthalmoscopy.

For this study myopia was defined [1, 3, 15, 16] as SER equal to or more minus than -0.50D, and hyperopia as SER greater than +0.50D. Thus, emmetropia was defined as SER in the range -0.49 to +0.50D. Only refractive data from right eyes was used for the current refractive class analysis as the correlation between right and left eye refraction was extremely high (P < 0.0005).

Continuous baseline variables were expressed as mean ± standard deviation and evaluated with a one-way analysis of variance. Categorical variables, such as sex were expressed as a percentage (%) and evaluated with the Chi-square test. The factors related to the prevalence of myopia, mean SER and astigmatism were assessed with univariate and multivariate regression analyses. The interaction test was used to assess the changing trend in the mean SER and age between males and females. A P-value < 0.05 was considered statistically significant. All analyses were performed using Empower (R) (www.empowerstats.com, X & Y solutions Inc., Boston, MA) and R (http://www.R-project.org).

## Results

Among the 4585 children who consented to participate in an eye examination, the data of 4365 (95.2%) children were used for analysis. Of the 220 children whose data were excluded (67, 67 and 86 cases in 2014, 2016 and 2018, respectively), 112 were excluded for strabismus (64 exotropia, 47 esotropia, 1 vertical strabismus), 89 were absent on the day of the eye examination or returned home early before the eye examination, 8 attended twice (but only data from the first attendance has been included), 11 were excluded due to pathology (2 cases of unilateral corneal opacity, 2 cases of bilateral severe keratoconus, 2 cases of pseudophakia following bilateral congenital cataract surgery, 2 cases of unilateral blindness due to congenital fundus abnormalities, 2 cases of congenital high myopia over -12D in both eyes, 1 case of nystagmus in conjunction with unilateral myopia of -20D). There were no significant differences in age and sex between the children whose data were analyzed and excluded (unpaired t-test, all P > 0.05). Table 1 shows a summary of the baseline data of the children included in the analysis. There were significant differences in sex and age between the three years of the study. Due to lower numbers of children in the younger (6 and 7 years) and older (14 and15 years) ends of the age range, data for the 6 and 7 years old were combined into one group and 14 and 15 were similarly combined for analysis. Two thirds of the children came from rural areas

**Table 1. Characteristics of the schoolchildren [mean ± SD or n (%)].**

| Group | All | 2014 | 2016 | 2018 | P-value (3 years) |
|---|---|---|---|---|---|
| **N** | 4365 | 1517 | 1394 | 1454 | |
| **Sex** | | | | | 0.01 |
| Male | 2054 (47.1%) | 696 (45.9%) | 629 (45.1%) | 729 (50.1%) | |
| Female | 2311 (52.9%) | 821 (54.1%) | 765 (54.9%) | 725 (49.9%) | |
| **Age** | 11.12 ± 1.66 | 11.08 ± 1.81 | 11.04 ± 1.66 | 11.23 ± 1.49 | 0.008 |
| **Age groups (N, % of total)** | | | | | < 0.001 |
| 6 and 7 | 158 (3.6%) | 74 (4.9%) | 57 (4.1%) | 27 (1.9%) | |
| 8 | 301 (6.9%) | 126 (8.3%) | 92 (6.6%) | 83 (5.7%) | |
| 9 | 552 (12.7%) | 189 (12.5%) | 202 (14.5%) | 161 (11.1%) | |
| 10 | 874 (20.0%) | 285 (18.8%) | 279 (20.0%) | 310 (21.3%) | |
| 11 | 1248(28.6%) | 408 (26.9%) | 390 (28.0%) | 450 (30.9%) | |
| 12 | 625 (14.4%) | 210 (13.8%) | 185 (13.3%) | 230 (15.8%) | |
| 13 | 408 (9.4%) | 128 (8.4%) | 134 (9.6%) | 146 (10.0%) | |
| 14 and 15 | 199 (4.6%) | 97 (6.4%) | 55 (3.9%) | 47 (3.2%) | |
| **Prevalence Refractive errors** | | | | | |
| Myopia | 176 (4.0%) | 53 (3.5%) | 61 (4.4%) | 62 (4.3%) | 0.41 |
| Hyperopia | 1935 (44.3%) | 959 (63.2%) | 528 (37.8%) | 448 (30.9%) | < 0.001 |
| Emmetropia | 2254 (51.6%) | 505 (33.3%) | 805 (57.8%) | 944 (64.9%) | < 0.001 |
| **Overall SER (D)** | 0.69 ± 0.91 | 0.89 ± 0.86 | 0.62 ± 0.89 | 0.56 ± 0.95 | < 0.001 |
| Male | 0.66 ± 0.88 | 0.84 ± 0.85 | 0.58 ± 0.81 | 0.56 ± 0.96 | < 0.001 |
| Female | 0.72 ± 0.94 | 0.93 ± 0.88 | 0.65 ± 0.96 | 0.56 ± 0.99 | < 0.001 |
| **Overall astigmatism (D)** | -0.16 ± 0.39 | -0.16 ± 0.40 | -0.17 ± 0.38 | -0.15 ± 0.39 | 0.32 |
| Male | -0.16 ± 0.40 | -0.16 ± 0.40 | -0.15 ± 0.39 | -0.15 ± 0.41 | 0.51 |
| Female | -0.16 ± 0.38 | -0.16 ± 0.37 | -0.17 ± 0.38 | -0.15 ± 0.39 | 0.43 |
| **Astigmatism† direction** | | | | | |
| WRT (30–0–150$^0$) | 474 (53.1%) | 131 (37.9%) | 146 (60.6%) | 197 (64.4%) | < 0.001 |
| ATR (60–120 $^0$) | 372 (41.7%) | 193 (55.8%) | 84 (34.9%) | 95 (31.0%) | < 0.001 |
| Oblique (31–59 $^0$ and 121–149 $^0$) | 47 (5.3%) | 22 (6.4%) | 11(4.6%) | 14 (4.6%) | 0.51 |

SER = spherical equivalent refraction.

† astigmatism > 0.25D.

(socioeconomic status level 5), one third came from low-density housing in the outer suburbs of Sydney [32].

The overall prevalence of myopia, emmetropia and hyperopia were 4.0%, 51.6% and 44.3%, respectively. With respect to age groups, myopia prevalence increased gradually from 2.5% to 7.0%, at age 6 and 7, and 14 and 15, respectively, with a corresponding increase in the prevalence of emmetropia from 38.0% to 54.8% (all P < 0.0001). For the same age range a decrease in prevalence of hyperopia was observed with age from 59.5% to 38.2% (P < 0.0001) (Table 2). Across the three time points, the prevalence of emmetropia increased from 33.3% to 57.8% and 64.9% in 2014, 2016 and 2018, respectively (P < 0.001). In comparison, over the 3 years, hyperopia prevalence decreased from 63.2% to 37.8% and 30.9%, respectively (P < 0.001), myopia prevalence was 3.5%, 4.4% and 4.3%, respectively. There was no significant difference in the prevalence of myopia between three years (all P > 0.05). Multivariate regression analyses showed that the older the age, the higher the prevalence of myopia (OR = 1.20, 95% CI: 1.09~1.31, P = 0.0001), but the prevalence of myopia was not related to sex (OR = 1.10, 95% CI: 0.81~1.49, P = 0.53) or year of testing (P > 0.05, Table 3).

**Table 2. Prevalence of refractive errors, mean Spherical Equivalent Refraction (SER) and astigmatism for all 4,365 schoolchildren according to sex and age.**

| | Prevalence (%) | | | Mean (D) | |
|---|---|---|---|---|---|
| | **Myopia** | **Emmetropia** | **Hyperopia** | **SER** | **Astigmatism** |
| **All** | 176 (4.0%) | 2254 (51.6%) | 1935 (44.3%) | 0.69 ± 0.91 | -0.16 ± 0.39 |
| **Sex** | | | | | |
| Male | 78 (3.8%) | 1109 (54.0%) | 867 (42.2%) | 0.66 ± 0.88 | -0.16 ± 0.40 |
| Female | 98 (4.2%) | 1145 (49.5%) | 1068 (46.2%) | 0.72 ± 0.94 | -0.16 ± 0.38 |
| *P* value | 0.46 | 0.003 | 0.008 | < 0.001 | 0.54 |
| **Age groups** | | | | | |
| **6 and 7** | 4 (2.5%) | 60 (38.0%) | 94 (59.5%) | 0.97 ± 1.04 | -0.15 ± 0.32 |
| **8** | 5 (1.7%) | 144 (47.8%) | 152 (50.5%) | 0.80 ± 0.84 | -0.14 ± 0.32 |
| **9** | 14 (2.5%) | 271 (49.1%) | 267 (48.4%) | 0.75 ± 0.86 | -0.16 ± 0.40 |
| **10** | 38 (4.3%) | 459 (52.5%) | 377 (43.1%) | 0.70 ± 0.96 | -0.15 ± 0.39 |
| **11** | 50 (4.0%) | 668 (53.5%) | 530 (42.5%) | 0.66 ± 0.83 | -0.18 ± 0.44 |
| **12** | 26 (4.2%) | 323 (51.7%) | 276 (44.2%) | 0.70 ± 1.02 | -0.17 ± 0.39 |
| **13** | 25 (6.1%) | 220 (53.9%) | 163 (40.0%) | 0.54 ± 0.82 | -0.16 ± 0.41 |
| **14 and 15** | 14 (7.0%) | 109 (54.8%) | 76 (38.2%) | 0.59 ± 1.07 | -0.17 ± 0.39 |
| **P value** | < 0.0001 | < 0.0001 | < 0.0001 | < 0.001 | 0.48 |

The overall mean SER was 0.69 ± 0.91D (range 9.88D to -5.75D), of which male and female means were 0.66 ± 0.88D and 0.72 ± 0.94D, respectively (P < 0.0001). The mean SER decreased significantly with age, from 1.00 ± 1.03D in the 6 and 7 years-old female cohort to 0.62 ± 1.30D in the 14 and 15 years-old cohort and from 0.92 ± 1.06D to 0.57 ± 0.73D in the corresponding male cohort. The mean SER was 0.89 ± 0.86D, 0.62 ± 0.89D and 0.56 ± 0.95D in 2014, 2016 and 2018, respectively (P < 0.001). Multivariate regression analyses showed that the mean SER was related to sex, age and year of testing. The mean SER of females was higher than that of males (β = 0.06, 95% CI: 0 ~ 0.11, P = 0.04). The mean SER demonstrates a significant shift towards less hyperopia with increasing age (β = -0.05, 95% CI: -0.06~ -0.03, P < 0.0001). Moreover, the mean SER in females (β = -0.06, 95% CI: -0.09~ -0.04, P < 0.0001) decreased faster than in males (β = -0.02, 95% CI: -0.05~ 0, P = 0.04) with age (P $_{interaction}$ = 0.02, Fig 1). The mean SER decreased slightly from 2014 to 2018 (P < 0.0001, Table 3).

The overall mean astigmatism was -0.16 ± 0.39D (range 0 to -5.00D). Seventy-nine percent of eyes exhibited negligible astigmatism (i.e. ≤0.25 D) and 18% showed astigmatism between

**Table 3. Multivariate regression analyses to estimate the changing trend of prevalence of myopia, mean Spherical Equivalent Refraction (SER) and astigmatism with years of test.**

| Exposure | Year | Non-adjusted, (95% CI) | P value | Model I, (95% CI) | P value | Model II, (95% CI) | P value |
|---|---|---|---|---|---|---|---|
| **Myopia prevalence** | 2014 | Reference | | Reference | | Reference | |
| | 2016 | 1.26 (0.87, 1.84) | 0.22 | 1.28 (0.88, 1.87) | 0.19 | 1.28 (0.88, 1.87) | 0.19 |
| | 2018 | 1.23 (0.85, 1.79) | 0.28 | 1.22 (0.84, 1.77) | 0.30 | 1.22 (0.84, 1.78) | 0.29 |
| **Mean SER** | 2014 | Reference | | Reference | | Reference | |
| | 2016 | -0.27 (-0.34, -0.20) | < 0.0001 | -0.27 (-0.34, -0.21) | < 0.0001 | -0.27 (-0.345, -0.21) | < 0.0001 |
| | 2018 | -0.32 (-0.39, -0.26) | < 0.0001 | -0.32 (-0.38, -0.25) | < 0.0001 | -0.31 (-0.38, -0.25) | < 0.0001 |
| **Mean astigmatism** | 2014 | Reference | | Reference | | Reference | |
| | 2016 | 0.03 (0, 0.06) | 0.11 | 0.03 (0, 0.06) | 0.11 | 0.03 (0, 0.06) | 0.12 |
| | 2018 | 0 (-0.03, 0.03) | 0.93 | 0 (-0.03,0.03) | 0.92 | 0 (-0.03, 0.03) | 0.93 |

Model I is adjusted for age; Model II is adjusted for age and sex.

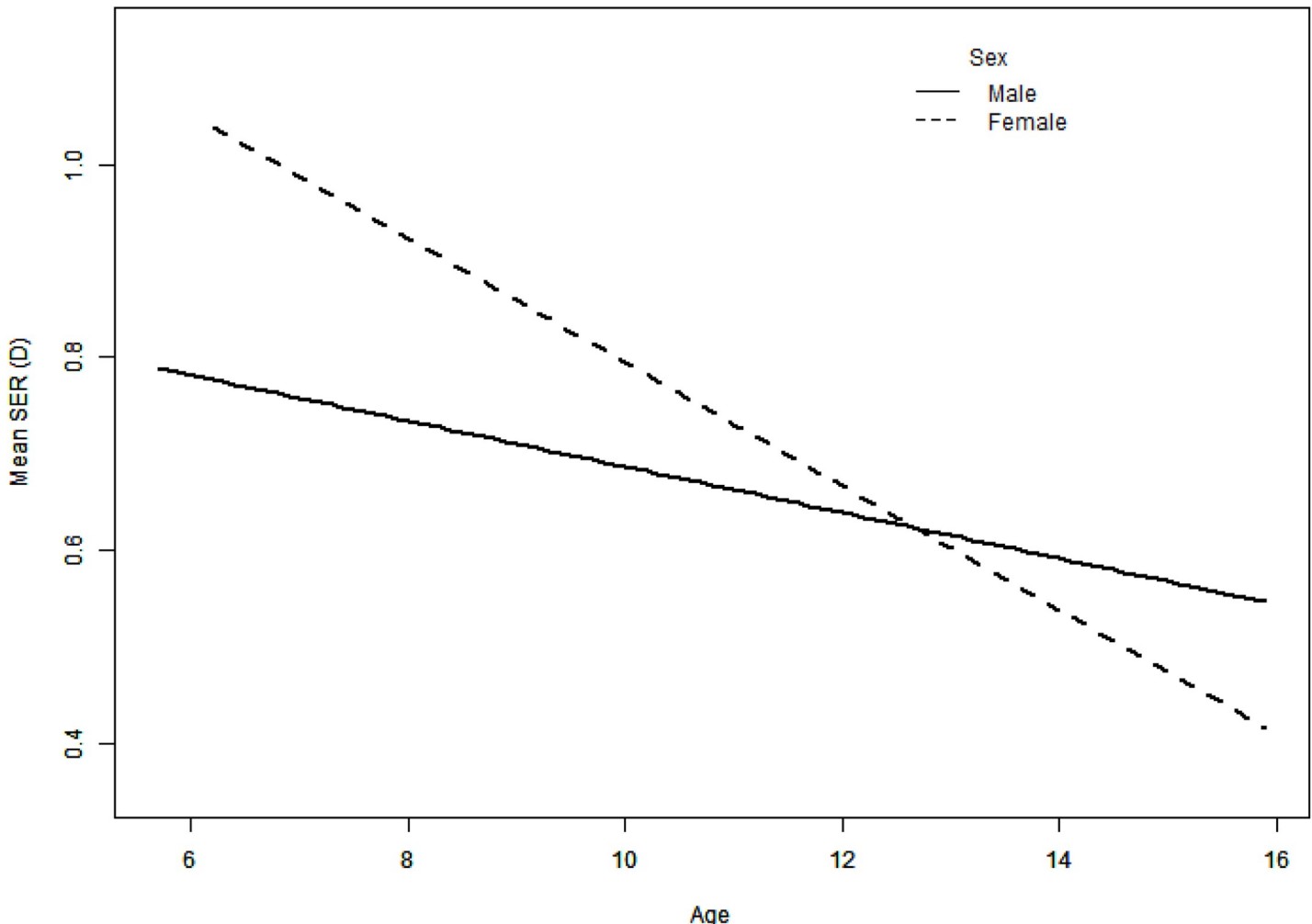

**Fig 1. Smooth curve fitting between mean spherical equivalent refraction (SER) and age.** Mean SER decreased with age both in males (solid lines, β = -0.02, 95% CI: -0.05~0, P = 0.04) and females (dashed lines, β = -0.06, 95% CI: -0.09~ -0.04, P < 0.0001). There was a faster mean SER reduction in females compared to males (P = 0.02, interaction test).

-0.25 D and -1.0 D. Multivariate regression analyses showed that the mean astigmatism was not related to sex (β = 0.002, 95% CI: -0.024~ 0.028, P = 0.89), age (β = -0.01, 95% CI: -0.02~ 0, P = 0.27) and year of testing (all P > 0.05). Of the 893 (21%) eyes with astigmatism >0.25 D in the current study, approximately 53% exhibited with-the-rule astigmatism (WTR: negative axis within 30˚ of the horizontal), nearly 5% exhibited oblique astigmatism (axis 30–60˚ or 120–150˚) and the remaining 42% exhibited against-the-rule astigmatism (ATR: axis 60– 120˚). Across the three time points, the prevalence of WTR increased from 37.9% to 60.6% and 64.4% in 2014, 2016 and 2018, respectively (P < 0.001). In comparison, over the 3 years, ATR prevalence decreased from 55.8% to 34.9% and 31.0%, respectively (P < 0.001), oblique astigmatism prevalence was 5.3%, 6.4% and 4.6%, respectively. (P = 0.51).

Compared with a similar population of children seen at Stewart House 40 years ago, and for data gathered using the same testing protocol (Table 4) [31], the age distribution was simi-lar, although the proportion of females was higher in the current study (53.1% vs. 48.6%). Using the definition of myopia (in either eye data, SER < 0) as used in the earlier report, the prevalence of myopia has doubled in both males and females (male: 7.4% vs. 3.3%; female:

**Table 4. Comparison between 1976 and the current study.**

| Study | 1976 | Current study (2014, 2016 and 2018) |
|---|---|---|
| N | 1166 (1 year) | 4365 (3 years) |
| Age: Min~Max (Mean) | 5~16 | 6~15 (11.13) |
| Sex (%): male / female | 51.4 / 48.6 | 46.9 / 53.1 |
| Prevalence (%): male / female | | |
| Hypermetropia † | 7.1 / 13.1 | 14.0 / 19.8 |
| Myopia † | 3.3 / 3.8 | 7.4 / 7.3 |
| Astigmatism † | | |
| Right eye | 7.9 / 11.6 | 7.1 / 7.4 |
| Left eye | 6.6 / 7.8 | 6.6 / 7.5 |

Definition criteria (Study in 1976: in either eye data. myopia = spherical equivalent refraction (SER) < 0; hypermetropia SER greater than >1.25D; astigmatism: >0.75D.
† Prevalence of hypermetropia (myopia or astigmatism) in males means percentage of males with hypermetropia (myopia or astigmatism) in the total number of males.

7.3% vs. 3.8%) and there is a higher prevalence hypermetropia greater than >1.25D (male: 14.0% vs. 7.1%; female: 19.8% vs. 13.1%) in the current study. The prevalence of astigmatism and sex differences were very similar across these two separate historic cohorts.

## Discussion

In this retrospective cross-sectional audit, we evaluated 4365 disadvantaged Australian schoolchildren and showed that myopia prevalence increased with age and mean SER decreased slightly from 2014 to 2018. Sex differences in the rate of change with age were observed. Compared with a similar population examined 40 years ago [31], the prevalence of myopia has doubled, but it remains significantly lower, and the refraction is slightly more hyperopic, than in other locations amongst schoolchildren of a similar age.

Other cross-sectional studies at different time points in different countries or areas also showed that the prevalence of myopia among schoolchildren increased over time (Table 5) [7,13,14,16, 33–36]. Differences in prevalence between studies may be related to race, age distribution, myopia definition and follow-up time in different studies. In these studies, the highest prevalence of myopia was in Asia (Mainland China, Hong Kong, Taiwan and Japan), second highest in United States and Israeli, third highest in Northern Ireland and Australian urban areas, and the lowest in disadvantaged Australian schoolchildren. The lower prevalence of myopia in Australia generally may be related to Australia's educational system and lifestyle [16]. This study has demonstrated for the first time, lower myopia prevalence and higher SER associated with socioeconomic disadvantage in Australia, particularly in a population of children from rural areas. Geographical remoteness is associated with children spending more physically active time outdoors than children living in urban areas and conceivably spending less free time on digital devices and near tasks. Educational expectations and learning outcomes are higher for children living in urban regions, particularly in high density housing [37–39].

In a study of Chinese schoolchildren in Hong Kong in 2005–2010 [5,6], the prevalence of myopia in schoolchildren aged 6 and 12, was similar to 20 years prior. In a study comparing the prevalence of myopia in 6- and 7-year- old children of Chinese ethnicity in Sydney and Singapore [40], the prevalence of myopia was significantly lower in Sydney than in Singapore. Taken together, these studies suggest that environmental factors such as living areas, lifestyles

**Table 5. Change in myopia prevalence and mean Spherical Equivalent Refraction (SER) of schoolchildren over time in different countries.**

| Author | Country | Study commenced (number) | Study completed (number) | Study duration (year) | Age (year) | Refraction method | Myopia definition | | Myopia prevalence (%) | | Mean SER (D) | |
|---|---|---|---|---|---|---|---|---|---|---|---|---|
| | | | | | | | | | From | To | From | To |
| **Chen et al [36] 2018** | Eastern China | 2001 (2418) | 2015 (2932) | 15 | 18.5 | Non—cycloplegic autorefraction | ≤ - 0.50 | all | 79.5 | 87.7 | - 2.5 | - 3.4 |
| **Lam et al [5] 2011** | Hong Kong Chinese | 1991 (383) | 2005–2010 (-) | 20 | 6–12 | Non—cycloplegic autorefraction | < - 0.50 | age 6 | 25 | 18.3 | - 0.03 | - 0.06 |
| | | | | | | | | age12 | 64 | 61.5 | - 1.45 | - 1.67 |
| **Lin et al [7] 2004** | Taiwanese | 1983 (4125) | 2000 (10878) | 18 | 7–18 | Cycloplegic autorefraction | < - 0.25 | age 7 | 5.8 | 20 | 0.52 | 0.17 |
| | | | | | | | | age 12 | 36.7 | 61 | - 0.48 | - 1.45 |
| | | | | | | | | age 15 | 64.2 | 81 | - 1.49 | - 2.89 |
| **Li et al [35] 2017** | Beijing, China | 2006 (3657) | 2015 (3676) | 10 | 15 | Cycloplegic autorefraction | < - 0.50 | all | 55.95 | 65.48 | - 2.23 | -3.13 |
| **Matsumura et al [33] 1999** | Japan | 1984 (-) | 1996 (-) | 13 | 3–17 | Non—cycloplegic autorefraction | ≤ - 0.50 | age 7 | 5 | 15 | 0.50 | 0.50 |
| | | | | | | | | age 12 | 35 | 60 | - 0.75 | - 1.75 |
| | | | | | | | | age 17 | 49.3 | 65.6 | - 1.75 | - 2.25 |
| **Vitale et al [13] 2009** | United States | 1971–1972 (-) | 1999–2004 (-) | 30 | 12–17 | Objective refraction or lensometry | < 0 | all | 24 | 33.9 | - | - |
| | | | | | | | | black | 12 | 31.2 | | |
| | | | | | | | | white | 25.8 | 34.5 | | |
| **Bar et al [34] 2005** | Israeli | 1990 (56639) | 2002 (83966) | 13 | 16–22 | Non—cycloplegic autorefraction | ≤ - 0.50 | all | 20.3 | 28.3 | - | - |
| **McCullough et al [17] 2016** | Northern Ireland | 2006–2008 (669) | 2011–2014 (212) | 6 | 12–13 | Cycloplegic autorefraction | ≤ - 0.50 | all | 16.4 | 14.6 | - 1.25 | - 1.25 |
| | | 1960 (-) | 2011–2014 (212) | 50 | | Cycloplegic autorefraction | ≤ - 0.50 | all | 7.2 (age 10–16) | 14.6 (age 12–13) | 1.8 (age 7) | 1.13 (age 7) |
| **French et al [14] 2013** | Sydney, urban | 2004–2005 (2353) | 2009–2011 (1084) | 6 | 12 | Cycloplegic autorefraction | ≤ - 0.50 | all | 13 | 14.4 | 0.38 | 0.31 |
| | | | | | | | | European | | | | |
| | | | | | | | | Caucasian | 4.4 | 8.3 | | |
| | | | | | | | | East Asian | 38.5 | 42.7 | | |
| **Junghans et al [16] 2005** | Sydney, urban | 1990 (2535) | 1998–2004 (1936) | 10 | 4–12 | Non—cycloplegic retinoscopy | ≤ - 0.50 | all | 6.5 | 8.4 | 0.50 | 0.45 |
| | | | | | | | | age 4 | 2 | 2.3 | | |
| | | | | | | | | age 12 | 10.9 | 14.7 | | |
| **Current study** | Sydney, disadvantaged | 2014 (1599) | 2018 (1499) | 5 | 6–15 | Non—cycloplegic retinoscopy | ≤ - 0.50 | all | 3.8 | 4.3 | 0.88 | 0.60 |
| | | | | | | | | age 7 | 1.4 | 1.7 | 1.09 | 0.94 |
| | | | | | | | | age 12 | 6.4 | 4.6 | 1.01 | 0.69 |
| | | | | | | | | age 15 | 6.8 | 6.4 | 0.62 | 0.58 |
| | | 1976 (1166) | 2014–2018 (4365) | 40 | 6–15 | Non—cycloplegic retinoscopy | ≤ - 0.50 | male | 3.3 | 4.9 | - | 0.76 |
| | | | | | | | | female | 3.8 | 5.4 | | 0.83 |

and early educational pressures may have an impact on the prevalence of myopia [40]. Our participants are from rural and outer suburban areas of New South Wales and the Australian

Capital Territory where Asian children are a minority and the ethnic mix of participants has not changed significantly over the period of the study [41].

In the current study, the mean SER showed a slight shift towards less hyperopia and the astigmatism remained stable from 2014 to 2018. This SER could be reasoned to be related to the significant reduction in the prevalence of hyperopia and increased prevalence of emmetropia, but stable prevalence of myopia over the three years. Several cross-sectional studies have also found at different time points that the refractions shifted in a myopic direction or became more severe myopic shift over time (Table 5) [7,14,17,33,35,36]. Compared with the studies above, the SER in this study showed a slightly higher degree of hyperopia for children of the same age. Moreover, the mean SER was hyperopic across all ages in the current study. The trend of changing refractive state with age was similar to a study of children from urban areas of Sydney [16], but the children in the latter study were younger (aged 4–12) and had a slightly lower hyperopic SER (0.60D vs. 0.45D). Compared with other studies, the schoolchildren with a slightly higher degree of hyperopia in this study may be partly responsible for the low and stable prevalence of myopia. Mean astigmatism was not related to age and sex. The trends of changing mean astigmatism with age [42,43] and sex [43] were similar to other studies.

There was no sex difference in the prevalence of myopia, although, females had a significantly higher mean hyperopic refraction than males in this study. These trends were consistent with that of children within a similar Australian population from 40 years ago [31]. In contrast, the later study of urban children in Sydney aged of 4-12-years-old found that there was no sex difference in myopia prevalence and mean refraction [16]. Studies in Beijing, China [44] and Taiwan [7] found that females had a higher prevalence of myopia and females had a higher mean refraction of myopia. In cross-sectional studies carried out in Beijing over a 10 year period [35] and in Western China over 15 years [36], the prevalence of myopia in females was higher than in males each year. However, a cross-sectional study of 6-12-year-olds in Hongkong [5] found that males and females had the same myopia prevalence, with older males having longer axial length and flatter corneal curvature. Contrasting findings in these prior studies may be related to different race and age distributions. Moreover, in the current study, we found that the mean SER of females decreased faster than males with age. Similar findings have been reported in both cross-sectional [44] and longitudinal studies [3]. This may be related to different visual experiences in daily life [35,44]. One possible explanation was that females tend to spend more time reading and performing near work and less time outdoors [39,45,46]. Children experiencing appropriate outdoor light intensity and those spending more time spent outdoors showed significantly less myopic shift and axial elongation [47,48].

The main strength of this study is that the cross-sectional data at 3 time points included 4365 disadvantaged Australian schoolchildren aged 6–15, and thus describes accurate and representative data on the changing trend of the related factors of myopia prevalence and mean SER. Potential limitations of non-cycloplegia in our analysis should be mentioned. Cycloplegic refraction is proposed as the gold standard for determining refractive error and yields better results than non-cycloplegic retinoscopy [49,50]. Non-cycloplegic retinoscopy can result in overestimation of myopia and underestimation of hyperopia in young children [50,51]. However, Yeotikar [52] found that mean difference in SER for 7 to 16 years children obtained by non-cycloplegic refraction with contralateral fogging and cycloplegic refraction was small (mean of 0.29D) and not clinically significant. Meanwhile, if we remove the 6–7 age group or 6–8 age group from the analysis, the overall trend of myopia prevalence with age does not change, indicating that this overestimation is unlikely to have occurred. In this study, the screening eye examination was part of a battery of other physical examinations, consequently cycloplegic refraction was not appropriate. In several previous Australian studies, non-cycloplegic refraction was performed in a similar age groups in a similar manner allowing a

meaningful comparison over time [15,16,31]. In addition, there are less children enrolled in the low-grade groups in 2018, with more children enrolled in the middle to high grade groups, comparing with 2014 and 2016 (ex. Only 1.9% in age group 6 and 7 in 2018 vs. 4.7% and 4.1% in 2014 and 2016, respectively). However, the School Principal selects all eligible children from disadvantaged families, so there is some variation in the age and sex profile of children year to year. Meanwhile, multivariate regression analyses accommodate these variations in numbers by adjusting for age and sex to estimate the changing trend of refractive error over time.

In conclusion, this study showed that the prevalence of myopia among disadvantaged Australian schoolchildren aged 6–15 has doubled compared with 40 years ago amongst schoolchildren of a similar age, but it continues to be significantly lower and the refraction is slightly more hyperopic than in other locations including in Australian urban areas. This may suggest that reasons for the large increase in prevalence of myopia reported in other countries must include questions relating to environmental risk factors, in particular, socioeconomic status, education and outdoor activity in addition to genetic propensity.

## Supporting information

**S1 Data.**
(XLSX)

## Acknowledgments

The authors wish to acknowledge Mr Graeme Philpotts, Chief Executive Officer of Stewart House and the staff of Stewart House for their support, and the families and children who participated in this study.

## Author Contributions

**Conceptualization:** Aicun Fu, Fiona Stapleton.

**Data curation:** Aicun Fu, Kathleen Watt, Barbara M. Junghans, Androniki Delaveris, Fiona Stapleton.

**Formal analysis:** Aicun Fu, Barbara M. Junghans, Fiona Stapleton.

**Funding acquisition:** Aicun Fu.

**Investigation:** Aicun Fu, Kathleen Watt, Fiona Stapleton.

**Methodology:** Aicun Fu, Androniki Delaveris, Fiona Stapleton.

**Project administration:** Aicun Fu.

**Resources:** Barbara M. Junghans.

**Software:** Aicun Fu, Fiona Stapleton.

**Supervision:** Fiona Stapleton.

**Validation:** Aicun Fu.

**Visualization:** Aicun Fu.

**Writing – original draft:** Aicun Fu.

**Writing – review & editing:** Aicun Fu, Kathleen Watt, Barbara M. Junghans, Fiona Stapleton.

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
