## [Decision Letter · Decision Letter 0]

27 Apr 2020

PONE-D-20-09893

Prevalence of myopia among disadvantaged Australian schoolchildren: a 5-year cross-sectional study

PLOS ONE

Dear Dr. Stapleton,

Thank you for submitting your manuscript to PLOS ONE. After careful consideration, we feel that it has merit but does not fully meet PLOS ONE’s publication criteria as it currently stands. Therefore, we invite you to submit a revised version of the manuscript that addresses the points raised during the review process.

We would appreciate receiving your revised manuscript by Jun 11 2020 11:59PM. To enhance the reproducibility of your results, we recommend that if applicable you deposit your laboratory protocols in protocols.io, where a protocol can be assigned its own identifier (DOI) such that it can be cited independently in the future. For instructions see: http://journals.plos.org/plosone/s/submission-guidelines#loc-laboratory-protocols

We look forward to receiving your revised manuscript.

Kind regards,

I-Jong Wang

Academic Editor

PLOS ONE

2. You indicated that you had ethical approval for your study and that parents provided consent for their children to attend the Stewart House service described in the study. In your Methods section, please ensure you have also stated whether you obtained consent specifically for participation in research from the parents or guardians of the minors included in the study or whether the research ethics committee or IRB specifically waived the need for their consent.

Reviewers' comments:

Reviewer's Responses to Questions

**Comments to the Author**

1. Is the manuscript technically sound, and do the data support the conclusions?

Reviewer #1: Yes

Reviewer #2: Yes

Reviewer #3: Yes

Reviewer #4: Yes

2. Has the statistical analysis been performed appropriately and rigorously? 

Reviewer #1: Yes

Reviewer #2: Yes

Reviewer #3: Yes

Reviewer #4: Yes

3. Have the authors made all data underlying the findings in their manuscript fully available?

Reviewer #1: Yes

Reviewer #2: Yes

Reviewer #3: Yes

Reviewer #4: Yes

4. Is the manuscript presented in an intelligible fashion and written in standard English?

Reviewer #1: Yes

Reviewer #2: Yes

Reviewer #3: Yes

Reviewer #4: Yes

5. Review Comments to the Author

Reviewer #1: It's a known fact, nothing new. Study of myopia,astigmatism separately,not using SER may be more important in the future. Full cycloplegic refraction data will be needed. Different age groups will be also needed to be separated.

Reviewer #2: This is a detailed study, but as you've mentioned, this study didn't use cycloplegic refraction. Since the overall spherical equivalent refraction was less than 1 diopter, the amount of refraction changed might be significant.

Reviewer #3: 1. In this study, refractive error was determined by noncycloplegic retinoscopy, instead of cycloplegic autorefractometry. Noncycloplegic refraction could result in overestimation of myopia and underestimation of hyperopia, especially in young children. Therefore, there would be some impacts in the statistical results. For example, in table 2, the higher prevalence of myopia in age group 6-7 than that in age group 8 could be the result of pseudomyopia in younger kids. In table 1, higher proportions of younger children (age 6-8) in 2014 group might result in more overestimation of myopia.

2. Children with ocular pathology were excluded from the study, but strabismus cases were not excluded. Without cycloplegia, the real refractive status could be masked by accommodation in strabismic children (ex. accommodative esotropia, intermittent exotropia ...).

3. The author did not mention that the retinoscopy was performed by one or multiple examiners. There would be inter-observer variability of retinoscopy.

4. The author did not mention the composition of subjects. As we know, race is an important factor of myopia prevalence.

5. There is no discussion about the significantly increased proportions of both myopia and hyperopia in this study as compared to the similar study in 1976.

Reviewer #4: The authors of the paper “Prevalence of myopia among disadvantaged Australian schoolchildren: a 5-year cross-sectional study “presents a cross-sectional analysis of refractive data and myopia prevalence in Australian children during the years 2014/2016/2018. The authors found that, although the prevalent of myopia remained low comparing with other locations, the mean spherical equivalent refraction (SER) decreased from 2014 to 2018, accompany with shifting towards less hyperopia. They also found that females had higher mean SER, but with faster decrease than male with age. In general, this paper is well-written and easy to follow. However, there are few points to be considered.

1.This paper found the mean SER decreased from 2016 to 2018, with trends of less hyperopia and stable myopia. However, reviewing the numbers of included cases, there are less children enrolled in the low grade groups in 2018, with more children enrolled in the middle to high grade groups, comparing with 2014 and 2016 (ex. Only 1.9% in age group 6 and 7 in 2018 vs. 4.7% and 4% in 2014 and 2016, respectively). Could the authors make some explanations that this will not lead to selection bias?

2.The authors found females had higher mean SER, but with faster decrease with age comparing with male. Unlike other studies with higher prevalence of myopia in female, the higher prevalence of hyperopia in female in the present study is somewhat different. This could possibly be explained by different visual experiences in daily life. However, similar to question 1, there are more female children included in this study, especially in 2014 and 2016, could this also be a confounding factor?

6. PLOS authors have the option to publish the peer review history of their article (what does this mean?). If published, this will include your full peer review and any attached files.

Reviewer #1: Yes: HON-LEUNG YUEN

Reviewer #2: No

Reviewer #3: No

Reviewer #4: No

---

## [Author Response · Author response to Decision Letter 0]

29 May 2020

Thank you for the opportunity to revise this manuscript for consideration. The comments have been carefully considered and a revised submission has been uploaded, including both a marked up version and clean version. Changes to the manuscript are highlighted in yellow and detailed responses to the reviewers’ comments are presented in italics below. 

Reviewer #1:

It's a known fact, nothing new. Study of myopia, astigmatism separately, not using SER may be more important in the future. Full cycloplegic refraction data will be needed. Different age groups will be also needed to be separated.

Answer: (1) Thank you for this comment. We have separated the astigmatic component as suggested and presented this by age (in Tables 1 and 2). 

We have added this information to lines 187-201, page 8: “The overall mean astigmatism was -0.16 ± 0.39D (range 0 to -5.00D). Seventy-nine percent of eyes exhibited negligible astigmatism (i.e. ≤0.25 D) and 18% showed astigmatism between -0.25 D and -1.0 D. Multivariate regression analyses showed that the mean astigmatism was not related to sex (β=0.002, 95% CI: -0.024~ 0.028, P =0.89), age (β= -0.01, 95% CI: -0.02~ 0, P = 0.27) and year of testing (all P > 0.05). Of the 893 (21%) eyes with astigmatism >0.25 D in the current study, approximately 53% exhibited with-the-rule astigmatism (WTR: negative axis within 30° of the horizontal), nearly 5% exhibited oblique astigmatism (axis 30 - 60° or 120 - 150°) and the remaining 42% exhibited against-the-rule astigmatism (ATR: axis 60 - 120°). Across the three time points, the prevalence of WTR increased from 37.9% to 60.6% and 64.4% in 2014, 2016 and 2018, respectively (P < 0.001). In comparison, over the 3 years, ATR prevalence decreased from 55.8% to 34.9% and 31.0%, respectively (P < 0.001), oblique astigmatism prevalence was 5.3%, 6.4% and 4.6%, respectively. (P = 0.51).”

 (2) This was a screening study and cycloplegic refraction was not determined. This limitation has been addressed in the discussion at lines 308-319, page12: “Cycloplegic refraction is proposed as the gold standard for determining refractive error and yields better results than non-cycloplegic retinoscopy. Non-cycloplegic retinoscopy can result in overestimation of myopia and underestimation of hyperopia in young children. However, if we remove the 6-7 age group or 6-8 age group from the analysis, the overall trend of myopia prevalence with age does not change, indicating that this overestimation is unlikely to have occurred. In this study, the screening eye examination was part of a battery of other physical examinations, consequently cycloplegic refraction was not appropriate. In several previous Australian studies, non-cycloplegic refraction was performed in a similar age groups in a similar manner allowing a meaningful comparison over time.”

Reviewer #2: 

This is a detailed study, but as you've mentioned, this study didn't use cycloplegic refraction. Since the overall spherical equivalent refraction was less than 1 diopter, the amount of refraction changed might be significant.

Answer: Please see the response to question 2, reviewer #1.

Reviewer #3: 

1. In this study, refractive error was determined by noncycloplegic retinoscopy, instead of cycloplegic autorefractometry. Noncycloplegic refraction could result in overestimation of myopia and underestimation of hyperopia, especially in young children. Therefore, there would be some impacts in the statistical results. For example, in table 2, the higher prevalence of myopia in age group 6-7 than that in age group 8 could be the result of pseudomyopia in younger kids. In table 1, higher proportions of younger children (age 6-8) in 2014 group might result in more overestimation of myopia.

Answer: (1) Noncycloplegic refraction may indeed result in overestimation of myopia and underestimation of hyperopia, especially in young children. We have further analysed the data based on age. In Table 2, there are 4(2.5%) and 5(1.7%) children with myopia in age 6-7 group and in age 8 group respectively. If we remove the 6-7 age group or 6-8 age group, the overall trend of myopia prevalence with age does not change. 

A sentence has been added at lines 312-314 as follows: “However, if we remove the 6-7 age group or 6-8 age group from the analysis, the overall trend of myopia prevalence with age does not change, indicating that this overestimation is unlikely to have occurred.”

In Table 1, there are 2(1%), 3(2%) and 4(3.6%) children with myopia aged 6-8 in 2014, 2016 and 2018, respectively. Although there are more children aged 6-8 in 2014, the proportion of myopia is not statistically higher. Meanwhile, we found that the mean SER decreased slightly and the mean astigmatism was stable from 2014 to 2018. 

2. Children with ocular pathology were excluded from the study, but strabismus cases were not excluded. Without cycloplegia, the real refractive status could be masked by accommodation in strabismic children (ex. accommodative esotropia, intermittent exotropia ...).

Answer: Thank you for this comment. We excluded the 112 strabismus cases (64 exotropia, 47 esotropia, 1 vertical strabismus) and reanalyzed the data. We added this content and changed all other relevant data throughout the manuscript. See lines 119-122, page 4: “the data of 4365 (95.2%) children were used for analysis. Of the 220 children whose data were excluded (67, 67 and 86 cases in 2014, 2016 and 2018, respectively), 112 were strabismus (64 exotropia, 47 esotropia, 1 vertical strabismus)”. Meanwhile, the conclusions of the article did not change after excluding the112 strabismus cases compared with including them.

3. The author did not mention that the retinoscopy was performed by one or multiple examiners. There would be inter-observer variability of retinoscopy.

Answer: Thank you for this comment. All the retinoscopy was performed by one examiner (AD). We added this content on lines 93-94, page 4.

4. The author did not mention the composition of subjects. As we know, race is an important factor of myopia prevalence.

Answer: Thank you for this comment. Please see the links (https://en.wikipedia.org/wiki/Asian_Australians) for specific data on Asians lived in Australia. The table "Metropolitan areas with significant Asian Australian populations (2016 Census)” seems to indicate that 97.7% of Asians live in the capital cities. Our participants are from rural and outer suburban areas and Asian children are a minority. This has been addressed in the discussion at lines 261-264(Our participants are from rural and outer suburban areas and Asian children are a minority and the ethnic mix of participants has not changed significantly over the period of the study).

5. There is no discussion about the significantly increased proportions of both myopia and hyperopia in this study as compared to the similar study in 1976.

Answer: The prevalence of myopia in Australia is increasing, consistent with reports from many other parts of the world. The main reason of the increased proportions of myopia may be the changing educational system and lifestyle. The slightly higher proportion of hyperopia compared to the earlier studies is consistent with another recent study in the Sydney urban population and is discussed at lines 272-279. 

Reviewer #4: 

The authors of the paper “Prevalence of myopia among disadvantaged Australian schoolchildren: a 5-year cross-sectional study “presents a cross-sectional analysis of refractive data and myopia prevalence in Australian children during the years 2014/2016/2018. The authors found that, although the prevalent of myopia remained low comparing with other locations, the mean spherical equivalent refraction (SER) decreased from 2014 to 2018, accompany with shifting towards less hyperopia. They also found that females had higher mean SER, but with faster decrease than male with age. In general, this paper is well-written and easy to follow. However, there are few points to be considered.

1. This paper found the mean SER decreased from 2016 to 2018, with trends of less hyperopia and stable myopia. However, reviewing the numbers of included cases, there are less children enrolled in the low grade groups in 2018, with more children enrolled in the middle to high grade groups, comparing with 2014 and 2016 (ex. Only 1.9% in age group 6 and 7 in 2018 vs. 4.7% and 4% in 2014 and 2016, respectively). Could the authors make some explanations that this will not lead to selection bias?

Answer: Thank you for this comment. The School Principal selects all eligible children from disadvantaged families, so there is some variation in the age and sex profile of children year to year. However, multivariate regression analyses accommodate these variations in numbers by adjusting for age and sex to estimate the changing trend of refractive error over time (Table 3). We have added this information to lines 319-327, page 12: “ In addition, there are less children enrolled in the low-grade groups in 2018, with more children enrolled in the middle to high grade groups, comparing with 2014 and 2016 (ex. Only 1.9% in age group 6 and 7 in 2018 vs. 4.7% and 4.1% in 2014 and 2016, respectively). However, the School Principal selects all eligible children from disadvantaged families, so there is some variation in the age and sex profile of children year to year. Meanwhile, multivariate regression analyses accommodate these variations in numbers by adjusting for age and sex to estimate the changing trend of refractive error over time.”

2.The authors found females had higher mean SER, but with faster decrease with age comparing with male. Unlike other studies with higher prevalence of myopia in female, the higher prevalence of hyperopia in female in the present study is somewhat different. This could possibly be explained by different visual experiences in daily life. However, similar to question 1, there are more female children included in this study, especially in 2014 and 2016, could this also be a confounding factor?

Answer: Thank you for this comment. As for the question regarding age, multivariate regression analyses accommodate these variations by adjusting for age and sex to estimate the changing trend of refractive error over time (Table 3).

---

## [Decision Letter · Decision Letter 1]

10 Jun 2020

PONE-D-20-09893R1

Prevalence of myopia among disadvantaged Australian schoolchildren: a 5-year cross-sectional study

PLOS ONE

Dear Dr. Stapleton,

Thank you for submitting your manuscript to PLOS ONE. After careful consideration, we feel that it has merit but does not fully meet PLOS ONE’s publication criteria as it currently stands. Therefore, we invite you to submit a revised version of the manuscript that addresses the points raised during the review process.

We look forward to receiving your revised manuscript.

Kind regards,

I-Jong Wang

Academic Editor

PLOS ONE

Reviewers' comments:

Reviewer's Responses to Questions

**Comments to the Author**

1. If the authors have adequately addressed your comments raised in a previous round of review and you feel that this manuscript is now acceptable for publication, you may indicate that here to bypass the “Comments to the Author” section, enter your conflict of interest statement in the “Confidential to Editor” section, and submit your "Accept" recommendation.

Reviewer #2: All comments have been addressed

Reviewer #3: All comments have been addressed

Reviewer #4: All comments have been addressed

2. Is the manuscript technically sound, and do the data support the conclusions?

Reviewer #2: Partly

Reviewer #3: Yes

Reviewer #4: Yes

3. Has the statistical analysis been performed appropriately and rigorously? 

Reviewer #2: Yes

Reviewer #3: Yes

Reviewer #4: Yes

4. Have the authors made all data underlying the findings in their manuscript fully available?

Reviewer #2: Yes

Reviewer #3: Yes

Reviewer #4: Yes

5. Is the manuscript presented in an intelligible fashion and written in standard English?

Reviewer #2: Yes

Reviewer #3: Yes

Reviewer #4: Yes

6. Review Comments to the Author

Reviewer #2: Thank you for our response. But I wonder that why did you choose to remove the 6-7 age group or 6-8 age

group from the analysis for supporting your opinion?

To my knowledge, previous study suggested that cycloplegia refraction should be performed before age 20, because the differences might up to 0.36±0.41D in the 13 year olds group (IOVS April 2010).

Reviewer #3: The analysis in this study is detailed. However, there are two weaknesses.

1. Noncycloplegic refractive examination was used.

2. There are a lot of similar studies in the literature, including quite a few studies of Australian children. This study did not provide anything new.

Reviewer #4: (No Response)

7. PLOS authors have the option to publish the peer review history of their article (what does this mean?). If published, this will include your full peer review and any attached files.

Reviewer #2: No

Reviewer #3: No

Reviewer #4: No

---

## [Author Response · Author response to Decision Letter 1]

29 Jul 2020

“Prevalence of myopia among disadvantaged Australian schoolchildren: a 5-year cross-sectional study”

We thank the reviewers again for their helpful suggestions and offer this rejoinder in response to the comments.

Reviewer #2: 

(1) I wonder that why did you choose to remove the 6-7 age group or 6-8 age group from the analysis for supporting your opinion? 

Response:

During the first round of comments on the manuscript, the third reviewer raised a question as follows ‘Noncycloplegic refraction potentially resulting in overestimation of myopia and underestimation of hyperopia, especially in young children. Therefore, there would be some impacts in the statistical results. For example, in table 2, the higher prevalence of myopia in age group 6-7 than that in age group 8 could be the result of pseudomyopia in younger kids. In table 1, higher proportions of younger children (age 6-8) in 2014 group might result in more overestimation of myopia.” To address this question, we reanalyzed the data with either the 6-7 age group or 6-8 age group removed. The overall trend of myopia prevalence with age did not change. This has been described at line 315 and we have made no further changes to the manuscript in this revision. 

(2) To my knowledge, previous study suggested that cycloplegia refraction should be performed before age 20, because the differences might up to 0.36±0.41D in the 13 year olds group (IOVS April 2010).

Response:

In the study referred to (IOVS April 2010),1 refractive error before and after cycloplegia was measured using a Humphrey 598 Autorefractor. They compared the difference between refractive error before and after cycloplegia resulting in the conclusion above. However, a different method was used in our study: refractive error was determined by non-cycloplegic retinoscopy using the fogging technique while the child maintained fixation on a distant (6 m) non-accommodative target. Yeotikar 2 found that mean difference in SER for 7 to 16 years children obtained by non-cycloplegic refraction with contralateral fogging and cycloplegic refraction was small (mean of 0.29D) and was not clinically significant. We have added this content to the section on limitations in the discussion (line 312-315).

 We agree that with respect to physiological aspects, the absence of cycloplegia (thus allowing accommodation as an uncontrolled variable) does affect one’s power to understand the degree of refractive latency to a certain degree. However, the dogmatic use of cycloplegics in all studies as dictated by the RESC protocol is not warranted, particularly in a screening examination in a respite environment as described in this study. The percentage of children affected by this latency is quite small and will have a very small impact on the mean SER of a cohort of thousands of children.

 Furthermore, the mandatory inclusion of cycloplegia avoids the issue arising from a number of reports alluding to the fact that cycloplegia can physiologically cause a significant minus shift in refractive error in a (albeit small) percentage of children (e.g. FA Young etal 1971, WM Ludlam et al 1972, HC Cheng & YT Hsieh 2014) and adults (TT Toh et al 2005). Other reports do not comment on this aspect, yet their data clearly indicate the same message when visually inspecting their scatter plots showing the difference in SER pre/post cycloplegic (e.g. R Fotedar etal 2007, where all dots below the x-axis represent children exhibiting a minus shift during cycloplegia). This minus-wards shift is due presumably to a forward shift of the crystalline lens during cycloplegia (L Gao etal 2002). Notably, the percentage of children shifting from one RESC-designated refractive group to another as a result of cycloplegia is very small, either plus-wards or minus-wards, indicating that net relative shift in numbers due to the physiological variable of accommodation does not necessarily mean that the any overall claims driven by the data will change substantially.

 As other non-cycloplegic studies also exist in the literature, we believe there are sufficient data for direct comparison with the results presented in this study and have not made further changes in the manuscript.

Reviewer #3: The analysis in this study is detailed. However, there are two weaknesses.

1. Noncycloplegic refractive examination was used.

Response: 

See above for Response to Reviewer 1, Q (2) 

2. There are a lot of similar studies in the literature, including quite a few studies of Australian children. This study did not provide anything new.

Response：

The other Australian data either represents a complete cross section of society (Sydney Myopia study data),3-5 or, comes mainly from higher socioeconomic groups (VEC).6,7 Our new data focusses more on lower socioeconomic groups, and the lower prevalence of myopia in those children from predominantly rural regions corresponds with the lower prevalence found in these social groups in other countries .

References

1. Paul G Sanfilippo, Byoung-Sun Chu, Olivia Bigault, et al. Up to What Age is a Cyclopleged Refraction Required? Results from the Twins Eye Study Tasmania (TEST). Acta Ophthalmol. 2014; 92(6): e458-62. 

2. Nisha S. Yeotikar, Ravi Chandra Bakaraju, P.S. Roopa Reddy, Kalyani Prasad. Cycloplegic refraction and non-cycloplegic refraction using contralateral fogging: a comparative study. Journal of Modern Optics. 2007;54(9):1317- 24.

3. French AN, Morgan IG, Burlutsky G, Mitchell P, Rose KA. Prevalence and 5- to 6-year incidence and progression of myopia and hyperopia in Australian schoolchildren. Ophthalmol. 2013;120(7):1482-91. 

4. French AN, O'Donoghue L, Morgan IG, Saunders KJ, Mitchell P, Rose KA. Comparison of refraction and ocular biometry in European Caucasian children living in Northern Ireland and Sydney, Australia. Invest Ophthalmol Vis Sci. 2012; 53(7):4021-31.

5. Elvis Ojaimi, Kathryn A Rose, Ian G Morgan, et al. Distribution of Ocular Biometric Parameters and Refraction in a Population-Based Study of Australian Children. Invest Ophthalmol Vis Sci. 2005;46(8):2748-54.

6. Junghans BM, Crewther SG. Prevalence of myopia among primary school children in eastern Sydney. Clin Exp Optom. 2003; 86(5):339-45. 

7. Junghans BM, Crewther SG. Little evidence for an epidemic of myopia in Australian primary school children over the last 30 years. BMC Ophthalmol. 2005; 5(1): 1.

---

## [Decision Letter · Decision Letter 2]

11 Aug 2020

Prevalence of myopia among disadvantaged Australian schoolchildren: a 5-year cross-sectional study

PONE-D-20-09893R2

Dear Dr. Stapleton,

We’re pleased to inform you that your manuscript has been judged scientifically suitable for publication and will be formally accepted for publication once it meets all outstanding technical requirements.

Kind regards,

I-Jong Wang

Academic Editor

PLOS ONE

Additional Editor Comments (optional):

Reviewers' comments:

Reviewer's Responses to Questions

**Comments to the Author**

1. If the authors have adequately addressed your comments raised in a previous round of review and you feel that this manuscript is now acceptable for publication, you may indicate that here to bypass the “Comments to the Author” section, enter your conflict of interest statement in the “Confidential to Editor” section, and submit your "Accept" recommendation.

Reviewer #2: All comments have been addressed

2. Is the manuscript technically sound, and do the data support the conclusions?

Reviewer #2: Yes

3. Has the statistical analysis been performed appropriately and rigorously? 

Reviewer #2: Yes

4. Have the authors made all data underlying the findings in their manuscript fully available?

Reviewer #2: Yes

5. Is the manuscript presented in an intelligible fashion and written in standard English?

Reviewer #2: Yes

6. Review Comments to the Author

Reviewer #2: Thank you for the detailed replication with addressing corresponding references. There was no further question about this article.

7. PLOS authors have the option to publish the peer review history of their article (what does this mean?). If published, this will include your full peer review and any attached files.

Reviewer #2: No

---

## [Editor Report · Acceptance letter]

17 Aug 2020

PONE-D-20-09893R2 

Prevalence of myopia among disadvantaged Australian schoolchildren: a 5-year cross-sectional study 

Dear Dr. Stapleton:

I'm pleased to inform you that your manuscript has been deemed suitable for publication in PLOS ONE. Congratulations! Your manuscript is now with our production department. 

Kind regards, 

on behalf of

Dr. I-Jong Wang 

Academic Editor

PLOS ONE